# Impacts of detritivore diversity loss on instream decomposition are greatest in the tropics

Luz Boyero [1,2✉], Naiara López-Rojo [1], Alan M. Tonin [3], Javier Pérez [1], Francisco Correa-Araneda [4], Richard G. Pearson [5,6], Jaime Bosch [7,8], Ricardo J. Albariño [9], Sankarappan Anbalagan[10], Leon A. Barmuta[11], Ana Basaguren [1], Francis J. Burdon [12], Adriano Caliman[13], Marcos Callisto [14], Adolfo R. Calor [15], Ian C. Campbell[16], Bradley J. Cardinale[17], J. Jesús Casas [18], Ana M. Chará-Serna [19,20], Eric Chauvet [21], Szymon Ciapała[22], Checo Colón-Gaud [23], Aydeé Cornejo [24], Aaron M. Davis[5], Monika Degebrodt[25], Emerson S. Dias[26], María E. Díaz[27,28], Michael M. Douglas [29], Andrea C. Encalada[30,39], Ricardo Figueroa [28], Alexander S. Flecker[31], Tadeusz Fleituch [32], Erica A. García [33], Gabriela García [34], Pavel E. García[35,36], Mark O. Gessner [25,37], Jesús E. Gómez[38], Sergio Gómez[31], Jose F. Gonçalves Jr [3], Manuel A. S. Graça [39], Daniel C. Gwinn [40], Robert O. Hall Jr [41], Neusa Hamada[42], Cang Hui [43,44], Daichi Imazawa[45], Tomoya Iwata [46], Samuel K. Kariuki[47], Andrea Landeira-Dabarca [30,40], Kelsey Laymon[23], María Leal[48], Richard Marchant[49], Renato T. Martins [42], Frank O. Masese [50], Megan Maul [51], Brendan G. McKie[12], Adriana O. Medeiros [15], Charles M. M' Erimba[47], Jen A. Middleton [29], Silvia Monroy[1], Timo Muotka[52], Junjiro N. Negishi[53], Alonso Ramírez [54], John S. Richardson [55], José Rincón[48], Juan Rubio-Ríos [18], Gisele M. dos Santos [14,56], Romain Sarremejane [52], Fran Sheldon[57], Augustine Sitati[50], Nathalie S. D. Tenkiano[58], Scott D. Tiegs [51], Janine R. Tolod [53], Michael Venarsky[57], Anne Watson[11] & Catherine M. Yule [59]

The relationship between detritivore diversity and decomposition can provide information on how biogeochemical cycles are affected by ongoing rates of extinction, but such evidence has come mostly from local studies and microcosm experiments. We conducted a globally distributed experiment (38 streams across 23 countries in 6 continents) using standardised methods to test the hypothesis that detritivore diversity enhances litter decomposition in streams, to establish the role of other characteristics of detritivore assemblages (abundance, biomass and body size), and to determine how patterns vary across realms, biomes and climates. We observed a positive relationship between diversity and decomposition, strongest in tropical areas, and a key role of abundance and biomass at higher latitudes. Our results suggest that litter decomposition might be altered by detritivore extinctions, particularly in tropical areas, where detritivore diversity is already relatively low and some environmental stressors particularly prevalent.

A full list of author affiliations appears at the end of the paper.

A key question in contemporary ecology is whether changes in biodiversity lead to alterations in the functioning of ecosystems and associated biogeochemical cycles[1,2]. Interest in this topic emerged in the 1990s, motivated in part by the remarkable increase in global biodiversity loss[3], and led to hundreds of experiments that manipulated biodiversity at different levels (species, genes or functional traits) in different groups of terrestrial and aquatic organisms, to examine possible effects on ecosystem processes[4,5]. While this large body of primary research and subsequent syntheses have demonstrated a strong, positive role of diversity of primary producers on biomass production[6–8], the patterns for decomposition have proven to be weaker and less consistent[6,9]. This contrast may occur because decomposition can be simultaneously affected by the diversities of plant litter, microbial decomposers and animal consumers, with consequently more complex relationships[10].

Plant litter decomposition is a key process in the biosphere, as 90% of the annual plant production escapes herbivory[11] and eventually becomes litter, which is ultimately decomposed or sequestered in terrestrial or aquatic ecosystems[10]. Streams play a particularly important role in receiving and processing litter from their catchments[12], contributing significantly to global carbon and nutrient fluxes[13–15]. Litter enters streams mainly in the form of leaves, and it is decomposed by microorganisms (mostly aquatic hyphomycetes) and specialised invertebrates (litter-consuming detritivores) that can obtain carbon and nutrients from the litter and associated fungi[16,17].

Multiple studies have manipulated detritivore diversity and assessed its effect on decomposition locally in streams or in laboratory microcosms, with inconsistent results[10]. These inconsistencies have been attributed to the existence of different species interactions driving either positive[18,19] or negative effects[20,21], which can compensate for each other and sometimes result in overall neutral effects[22]. However, there has been no global assessment of the relationship between detritivore diversity and decomposition in streams, which would help account for local and regional environmental contingencies in the diversity–decomposition relationship[23]. A meta-analysis of terrestrial and aquatic studies revealed strong effects of

detritivore diversity on decomposition, but there was no separate assessment of instream decomposition[9]. Several stream studies have suggested a direct link between faster decomposition[24] and greater detritivore diversity[25,26] in temperate streams, but did not explore the relationship explicitly. A large-scale study demonstrated that decomposition in streams was enhanced when detritivore assemblages were more complex (large- and medium-sized organisms as opposed to medium-sized only), although it did not examine detritivore diversity[27].

Here, we describe results from a global-scale decomposition experiment conducted by partners of the GLoBE collaborative research network (www.globenetwork.es) in 38 streams distributed across 23 countries in all inhabited continents. We use a standardised design and methodology to examine global-scale ecological questions, which reduces the number of confounding factors that need to be statistically controlled for in a meta-analysis[28,29]. Our main working hypothesis is that detritivore diversity has a major positive effect on decomposition[9], although we also expect an influence of other detritivore assemblage characteristics such as abundance, biomass, and body size[18,22,27]. Moreover, we predict that biotic drivers of decomposition vary across sites at different latitudes, possibly because of the varying interplay between positive and negative species interactions[22]. We also explore detritivore variation across latitudes, biogeographic realms, biomes and climates, to further explain their global distribution and the potential consequences of reduced diversity for decomposition in different areas of the world. Unlike previous large-scale decomposition studies using 1 or 2 litter types[24,30], we use several mixtures representing a variety of litter traits to maximise the generality of our results. Our global experiment supports the expected positive relationship between detritivore diversity and decomposition, and reveals that detritivore species loss may have its greatest consequences on stream ecosystem functioning in the tropics.

## Results

The model that best explained global variation in total decomposition explained 73% of the variation and revealed a significant influence of detritivore diversity, abundance, biomass, latitude, and interactions between diversity and latitude, abundance and latitude, and biomass and latitude (Table 1 and Supplementary Table 1). The model that best explained global variation in detritivore-mediated decomposition explained 82% of variation in the data, and showed that the interactions between diversity and latitude, abundance and latitude, and biomass and latitude were significant (Table 1 and Supplementary Table 1). As these results indicated that the three detritivore variables were important predictors of decomposition, but their influence varied with latitude, we explored the interactions with a second type of model where latitude was a categorical variable (Supplementary Table 2). These models revealed that the relationship between detritivore diversity and decomposition was stronger in tropical areas than in temperate areas and absent in boreal areas; and that abundance and biomass were important in temperate and boreal areas, but not in tropical areas (Fig. 1 and Supplementary Table 2).

All detritivore variables varied significantly among realms, biomes and climates, and so did assemblage composition (Figs. 2–4, Table 2 and Supplementary Table 3). Diversity and abundance were highest in the Palearctic realm, tundra and temperate broadleaf and coniferous forests, and warm temperate and snow climates; and lowest in Neotropical, Afrotropical and Indomalayan realms, tropical wet forests and savannas and xeric shrublands, and equatorial climates. Biomass and mean body size were highest in Palearctic and Nearctic realms, temperate broadleaf and coniferous forests, and again warm temperate and snow climates, with the lowest values in the Indomalayan realm, tropical savannas and xeric shrublands, and

**Table 1 Results of the best additive models explaining variation in total and detritivore-mediated litter decomposition based on detritivore diversity, abundance, biomass, mean body size, latitude, and interactions between detritivore variables and latitude.**

| Effect | edf | F | p |
|---|---|---|---|
| Total decomposition | | | |
| Diversity | 4.00 | 6.94 | <0.001 |
| Abundance | 3.14 | 6.34 | <0.001 |
| Biomass | 1.00 | 2.00 | 0.159 |
| Mean body size | 1.86 | 2.10 | 0.102 |
| Latitude | 1.00 | 3.01 | 0.085 |
| Diversity × latitude | 14.56 | 6.17 | <0.001 |
| Abundance × latitude | 1.00 | 8.67 | 0.004 |
| Biomass × latitude | 7.91 | 4.20 | <0.001 |
| Detritivore-mediated decomposition | | | |
| Diversity | 4.00 | 0.53 | 0.716 |
| Abundance | 1.05 | 0.01 | 0.912 |
| Biomass | 1.00 | 0.04 | 0.843 |
| Mean body size | 1.08 | 1.00 | 0.843 |
| Latitude | 1.71 | 0.27 | 0.763 |
| Diversity × latitude | 14.14 | 4.74 | <0.001 |
| Abundance × latitude | 8.76 | 3.30 | <0.001 |
| Biomass × latitude | 7.99 | 4.36 | <0.001 |

All predictors were fitted as tensor product interaction smooths. We show effective degrees of freedom (edf) and values of F and p for each factor. Models explained 69% and 78% of variation in the data, respectively.

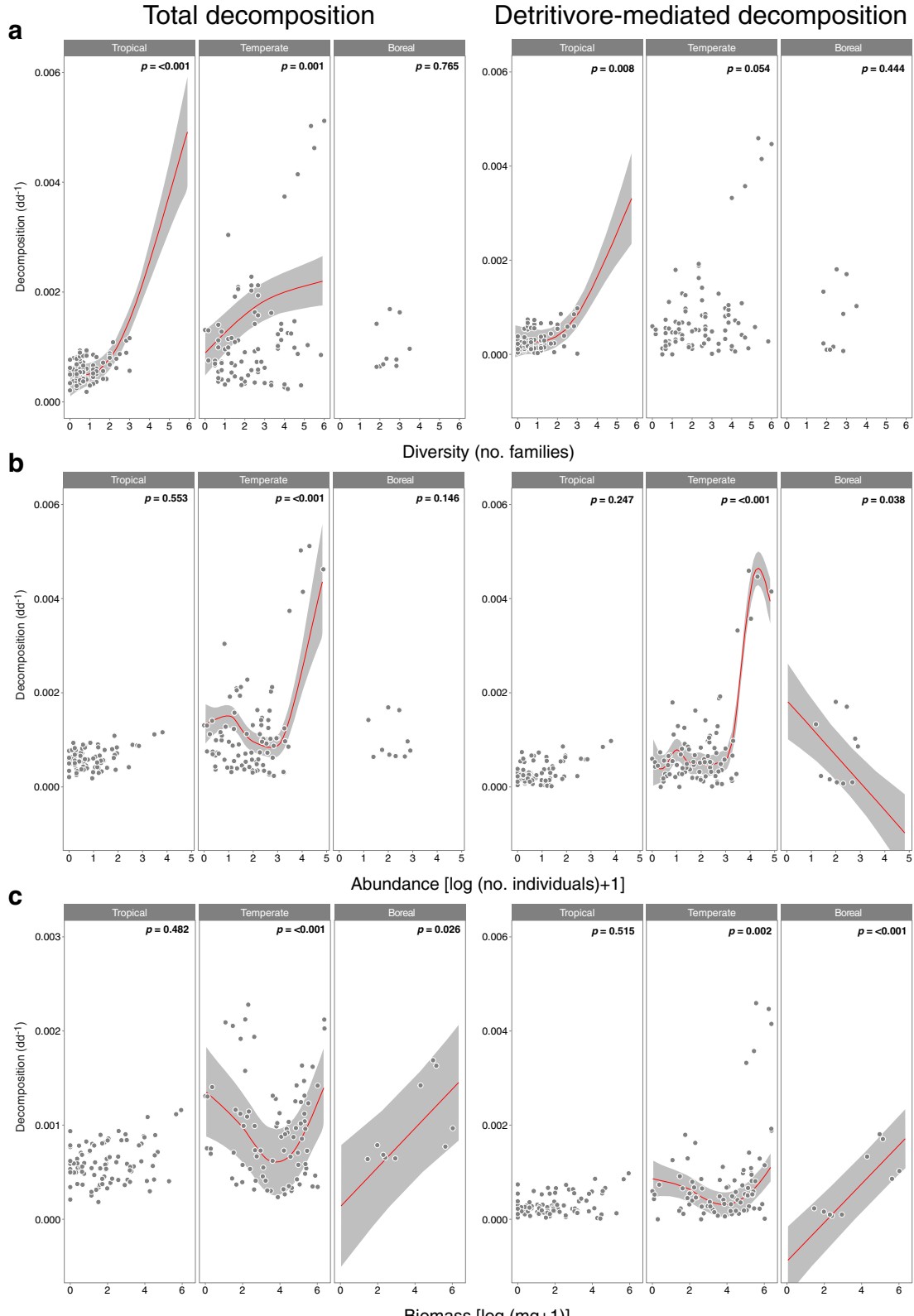

**Fig. 1 Generalised additive models exploring the influence of detritivore diversity, abundance and biomass on decomposition in different latitudinal zones (tropical: ≤23°; temperate: 24–60°; and boreal: >60°).** Variation in total and detritivore-mediated decomposition (measured as the proportion of litter mass loss per degree day, dd; mean ± SE) with **a** detritivore diversity (number of families per litterbag), **b** log-transformed abundance (number of individuals per litterbag) and **c** log-transformed biomass (mg per litterbag), in different latitudinal zones. Lines represent the smoothers and shading the 95% confidence intervals from generalised additive models for significant relationships (p-value < 0.05); whole-model results are given in Supplementary Table 3.

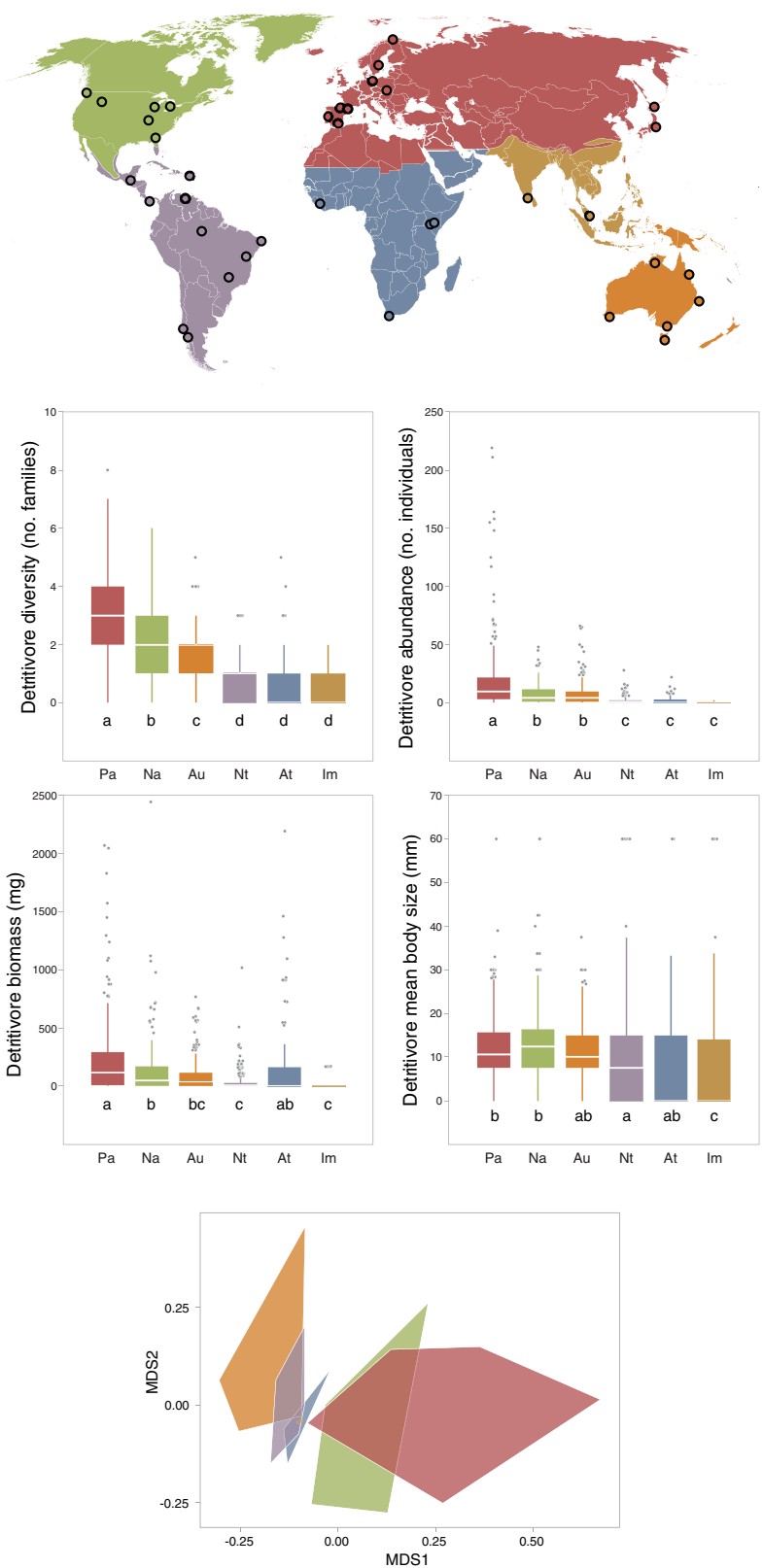

**Fig. 2 Global distribution of study sites in different biogeographic realms (Pa, Palearctic; Na, Nearctic; Au, Australasian; Nt, Neotropical; At, Afrotropical; Im, Indomalayan); *n* = 38.** Box plots show the median, interquartile range and minimum-maximum range of litter-consuming detritivore diversity (number of families per litterbag), abundance (number of individuals per litterbag), biomass (mg per litterbag) and mean body size (mm) in each realm (ordered from highest to lowest diversity); different letters indicate significant differences. The NMDS ordination of litter-consuming detritivores with realms is represented by polygons of different colours as in maps and box plots. Significant differences in assemblage structure were: Pa vs. Na, At, Au, Im; Na vs. Nt, Au; Nt vs. Au.

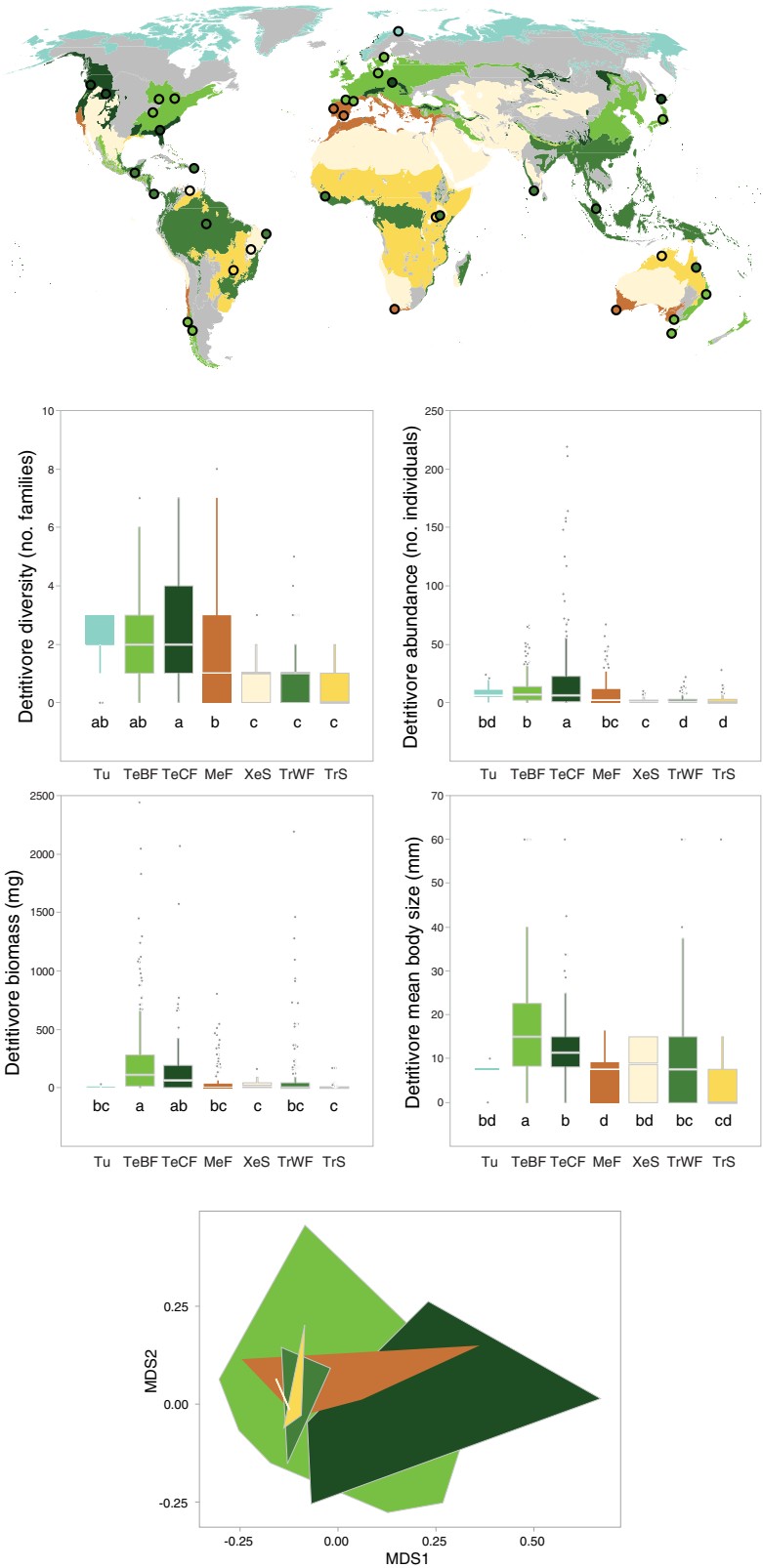

**Fig. 3 Global distribution of study sites in different biomes (Tu, tundra; TeBF, temperate broadleaf forest; TeCF, temperate coniferous forest; MeF, Mediterranean forest; XeS, xeric shrubland; TrWF, tropical wet forest; TrS, tropical savanna); _n_ = 38.** Box plots show the median, interquartile range and minimum-maximum range of litter-consuming detritivore diversity (number of families per litterbag), abundance (number of individuals per litterbag), biomass (mg per litterbag) and mean body size (mm) in each biome (ordered from highest to lowest diversity); different letters indicate significant differences. The NMDS ordination of litter-consuming detritivores with biomes is represented by polygons of different colours as in maps and box plots. Significant differences in assemblage structure were: TrWF vs. TeBF, TeCF, MeF.

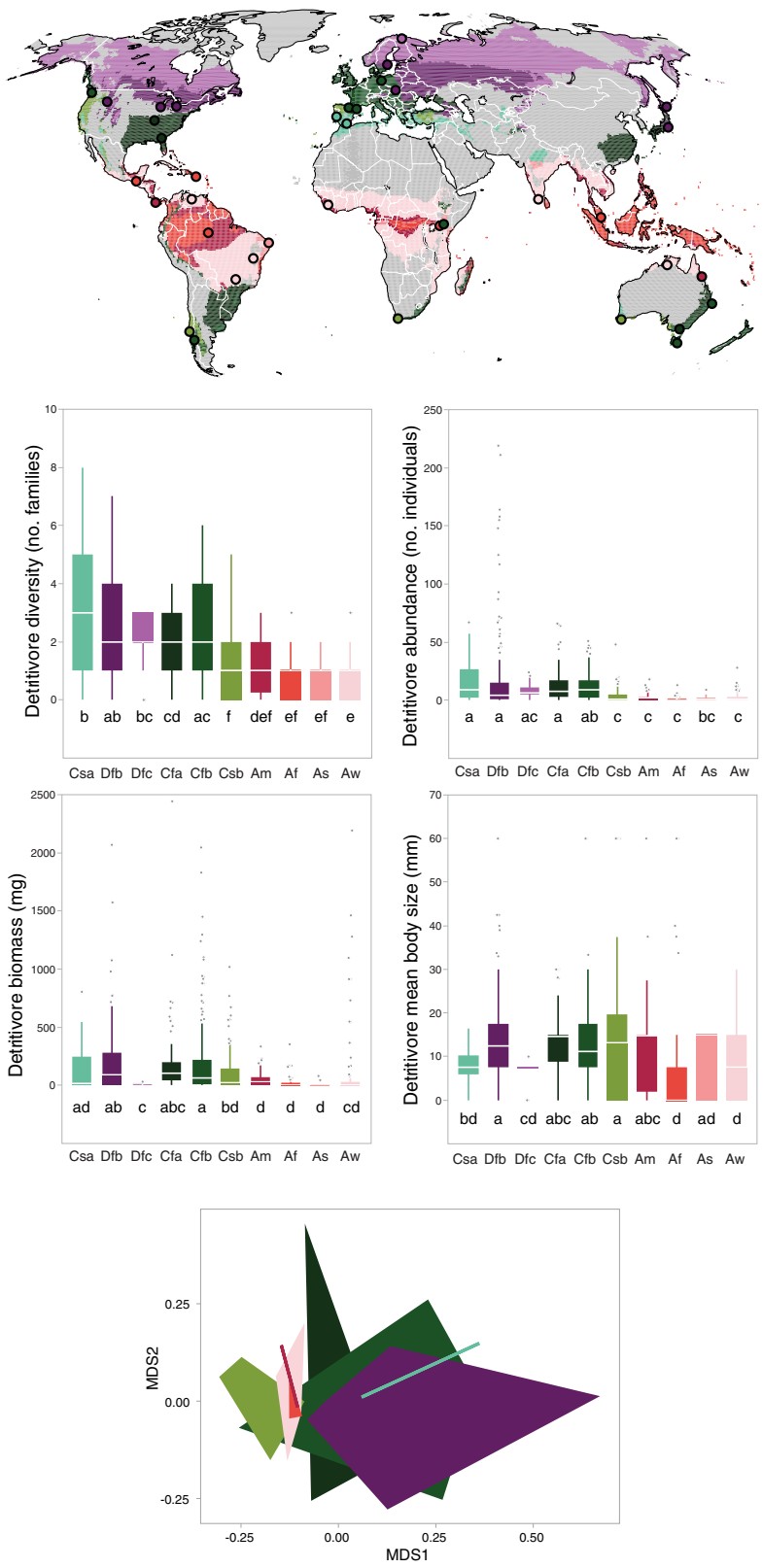

equatorial climates. Assemblage composition mostly differed between the Palearctic/Nearctic (with many families of Laurasian origin) and other realms (families of Gondwanan distribution); between tropical wet forests and several other biomes; and between equatorial and other climates.

## Discussion

Our study demonstrates a positive influence of detritivore diversity on decomposition, supporting previous suggestions that latitudinal gradients in detritivore diversity and instream decomposition are linked[24,25] and agreeing with results of a

**Fig. 4 Global distribution of study sites in different climates [A, equatorial (Af, fully humid; Am, monsoon; As, with dry summer; Aw, with dry winter); C, warm temperate (Cfa, fully humid with hot summer; Cfb, fully humid with warm summer; Csa, with dry and hot summer; Csb, with dry and warm summer); D, snow (Dfb, fully humid with warm summer; Dfc, fully humid with cold summer)]; _n_ = 38.** Box plots show the median, interquartile range and minimum-maximum range of litter-consuming detritivore diversity (number of families per litterbag), abundance (number of individuals per litterbag), biomass (mg per litterbag) and mean body size (mm) in each climate (ordered from highest to lowest diversity); different letters indicate significant differences. The NMDS ordination of litter-consuming detritivores with biomes is represented by polygons of different colours as in maps and box plots. Significant differences in assemblage structure were: Aw vs. Cfb, Cfa, Dfb; Af vs. Cfa, Cfb, Dfb.

---

**Table 2 Results of linear mixed effects models exploring variation in detritivore and total invertebrate diversity, abundance, biomass and mean body size, and PERMANOVAs exploring variation in assemblage composition, among realms, biomes and climates.**

| Effect | df | F | p |
|---|---|---|---|
| Diversity | | | |
| Realms | 6, 1090 | 387.33 | <0.001 |
| Biomes | 7, 1089 | 251.67 | <0.001 |
| Climates | 10, 1086 | 196.78 | <0.001 |
| Abundance | | | |
| Realms | 6, 1090 | 109.38 | <0.001 |
| Biomes | 7, 1089 | 64.70 | <0.001 |
| Climates | 10, 1086 | 58.46 | <0.001 |
| Biomass | | | |
| Realms | 6, 1090 | 44.16 | <0.001 |
| Biomes | 7, 1089 | 60.57 | <0.001 |
| Climates | 10, 1086 | 31.64 | <0.001 |
| Mean body size | | | |
| Realms | 6, 1090 | 472.25 | <0.001 |
| Biomes | 7, 1089 | 472.33 | <0.001 |
| Climates | 10, 1086 | 363.65 | <0.001 |
| Composition | | | |
| Realms | 5, 37 | 2.30 | 0.002 |
| Biomes | 6, 37 | 1.54 | 0.015 |
| Climates | 9, 37 | 1.32 | 0.029 |

We show degrees of freedom (df) for numerator and denominator, and values of _F_ and _p_ for each factor. Realms: Pa, Palearctic; Ne, Nearctic; Au, Australasian; Nt, Neotropical; At, Afrotropical; and In, Indomalayan. Biomes: Tu, tundra; TeBF, temperate broadleaf forest; TeCF, temperate coniferous forest; MeF, Mediterranean forest; XeS, xeric shrubland; TrWF, tropical wet forest; and TrS, tropical savanna. Climates: A, equatorial (Af, fully humid; Am, monsoon; As, with dry summer; Aw, with dry winter); C, warm temperate (Cfa, fully humid with hot summer; Cfb, fully humid with warm summer; Csa, with dry and hot summer; Csb, with dry and warm summer); D, snow (Dfb, fully humid with warm summer; Dfc, fully humid with cold summer).

meta-analysis of controlled experiments performed in terrestrial and aquatic ecosystems[9]. Our result also agrees with results of controlled experiments that found average increases in decomposition of 12–30% per detritivore species added[18,19,31], suggesting that positive interactions (i.e. resource partitioning and facilitation) are prevalent in detritivore assemblages. Clearly, our field study does not demonstrate causality among these variables or the suggested mechanisms, but the finding of a consistent relationship across 113° of latitude indicates that detritivore diversity, at least at the family level, is indeed a driver of decomposition. Whether this relationship would change by considering species diversity cannot be currently ascertained due to limited taxonomic knowledge in many regions[32].

The relationship between detritivore diversity and decomposition, when data were grouped according to latitudinal zone, was most evident in tropical areas, less important in temperate areas and unimportant in boreal areas (although the latter were underrepresented in our dataset). Others have demonstrated a positive relationship between detritivore diversity and decomposition in some streams of boreal areas[33], but our global dataset indicates a relatively weak relationship when compared to other latitudinal zones. Importantly, the stronger relationship between detritivore diversity and decomposition in the tropics suggests that species losses in these areas, where detritivore diversity is already lower than at higher latitudes as shown here and elsewhere[25,26], may cause the greatest impact on decomposition. Detritivores in tropical areas are particularly vulnerable, because of the prevalence of multiple environmental stressors. For example, concentrations of agricultural pesticides have limited regulation in many tropical countries[34] and are known to cause mortality in many detritivores[35–37]. Climate warming is also likely to cause more extinctions in the tropics because more detritivore species are closer to their thermal maxima than elsewhere[25] and are likely to suffer greater physiological changes, despite the smaller changes in temperature occurring in this latitudinal zone[38]. Nevertheless, other climatic changes such as increased droughts can be more important at higher latitudes[39].

We found that the influence of detritivore abundance and biomass on decomposition also varied with latitude, but with negligible effect in the tropics and more important at higher latitudes. These variables have previously been found to be important predictors of decomposition in some tropical streams[40], but here their importance was lower in the tropics than elsewhere. In temperate areas, both relationships were non-linear and complex (with decomposition first decreasing and then increasing with higher abundance or biomass), which impedes predictions about how decomposition might be altered by changes in these variables. Moreover, responses of abundance and biomass to environmental stressors are not as straightforward as diversity loss, because lost species can be replaced by more tolerant ones that thrive under stressful conditions and can cause an overall increase in numbers[41,42]. Smaller detritivores are often more sensitive to stressors than larger ones[42], although this variation could be due to taxonomic differences rather than to size. Our results suggest that species replacements under environmental stress could result in an overall increase in biomass, but this possibility needs confirmation.

The distribution of most detritivore families corresponded to broad realms (Fig. 5), with 26 families showing a Laurasian distribution (i.e. being present in the Palearctic and/or Nearctic realms) and 14 families a Gondwanan distribution (Neotropical, Afrotropical, Australasian, and/or Indomalayan realms). Although we did not perform phylogenetic analyses, this dichotomy, together with the observation that diversity and abundance of detritivores were higher in the Palearctic and Nearctic (and their predominant biomes and climates), suggests that patterns of variation in diversity and abundance were at least partly determined by biogeography. Our findings contrast with those for angiosperms, current distributions of which do not correspond to tectonic history, possibly because of the existence of high transoceanic dispersal[43]; however, they support patterns for organisms with lower dispersal, such as liverworts and conifers[44], which show clear Laurasian–Gondwanan disjunctions[45].

The strong influence of biogeography on detritivore diversity and abundance, and the fact that these two variables are key

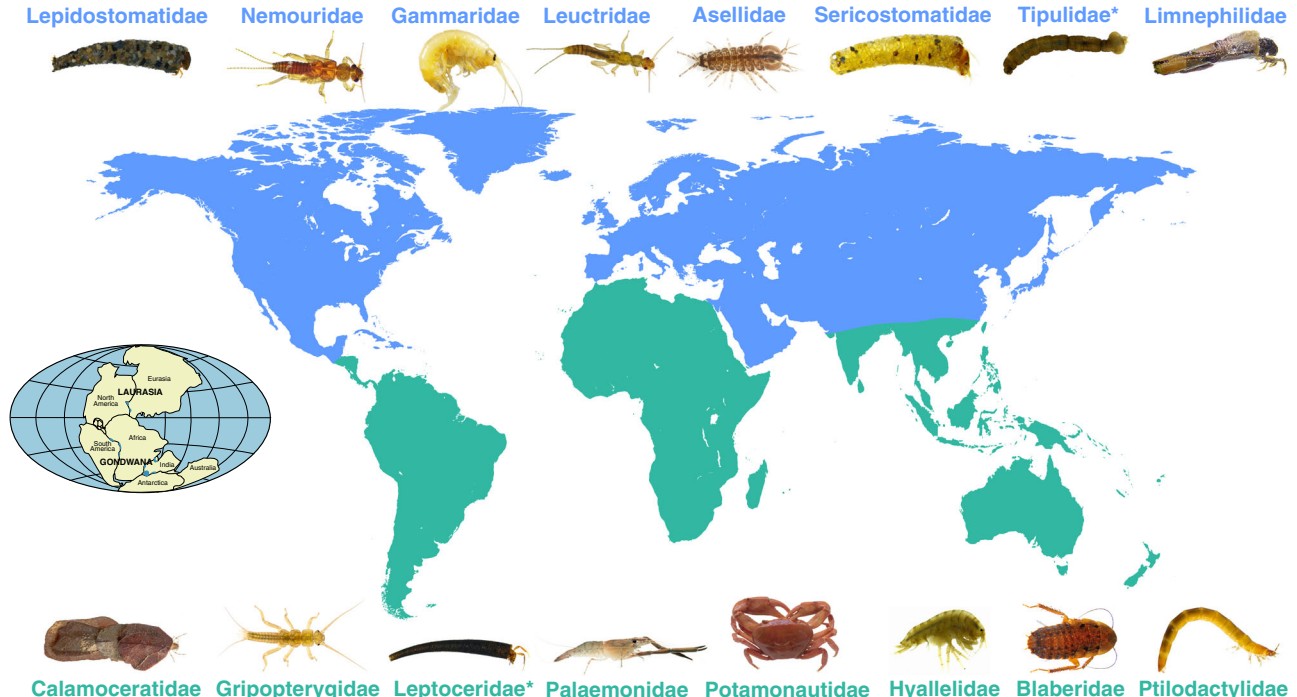

**Fig. 5 Distribution of detritivore families in our study, which was predominantly Laurasian (blue) or Gondwanan (green); insert indicates origins of those two regions (≈200 Ma).** Photographs represent a subset of families (ordered left to right from the most to the least abundant in our study) and asterisks denote families that were globally distributed but more abundant in one of the two areas. A complete list of families is provided in Supplementary Table 1. Photograph credits: L. Boyero, A. Cornejo, R. Figueroa, N. López-Rojo, F. Masese, J. Pérez, J. Rubio-Ríos, J. Vieira and C. M. Yule.

drivers of decomposition, suggest that the split of Pangea in the Late Jurassic (≈200 Ma ago) had a crucial legacy effect on the current functioning of stream ecosystems and the influence of ongoing environmental change. The lower detritivore diversity of tropical streams[25] and the higher susceptibility of their fauna to extinction[38] make these streams more vulnerable to reductions in decomposition rates that are associated with impaired ecosystem functioning[46,47]. This observation, together with the over-exploitation of natural resources that severely affects tropical stream ecosystems[48], indicates that tropical detritivore species should be of high conservation concern globally.

## Methods

**Study sites**. We conducted our study in 38 headwater streams located in different regions in 23 countries (Figs. 2–4). A random distribution of sites was unfeasible, so some regions were underrepresented (mostly Africa and northern Asia), which is usually the case for globally distributed experiments[28,49,50]. Streams were similar in size (mean ± SE: wetted channel width, 3.9 ± 0.1 m; water depth, 28.7 ± 0.4 cm; 1st–3rd order) and physical habitat (alternating riffles and pools). Most had rocky substrate and were shaded by a dense riparian vegetation (64 ± 1%) representative of the region. They were located in 6 realms, 7 biomes, and 10 Köppen climate classes[51]. In each stream we selected a ca. 100-m long reach with 5 consecutive pool habitats in which to conduct the experiment. Further information on site physi-cochemical characteristics is given in Supplementary Table 4.

**Field and laboratory work**. At each site, we incubated 6 different 3-species litter mixtures, which included 9 species in total (Supplementary Table 5). The species and mixtures were chosen to represent different levels of functional diversity for a companion study[52], but here our interest was to use a variety of mixtures and thus increase the generality of our results (as opposed to working with a single or a few species). The 9 species were collected at different locations around the world and distributed among partners[52]; we considered the possible home-field-advantage effect of using litter from different origins negligible based on available literature[53,54].

Litter mixtures were enclosed within paired coarse-mesh (5 mm) and fine-mesh (0.4 mm) litterbags containing the same amount and type of litter. The two types of litterbag respectively quantified total and microbial decomposition, and allowed calculation of detritivore-mediated decomposition (see below). There were 60

litterbags per stream ($n = 5$ per litter mixture and mesh size), each containing 3 g of senescent litter (1 g per species), which had been collected freshly fallen from the forest floor, air-dried and distributed among research partners[52]. Litterbags were deployed in each stream (one litterbag per litter mixture type and mesh size in a different stream pool, with all 5 pools consecutive) in 2017–2019 at the local time of the year with the greatest litter input and were retrieved after 23–46 d, depending on water temperature in each stream, thereby halting the decomposition process at a comparable stage (mean ± SD: 32 ± 17% litter mass loss on average for all the litter mixtures, 41 ± 18% for the fastest decomposing mixture[52]; mean values for each biome are given in Supplementary Fig. 1). Litterbags were transported to the laboratory on ice enclosed individually in zip-lock bags and rinsed with filtered stream water to remove attached sediment and invertebrates. Litter was oven-dried (70 °C, 72 h) and a subsample weighed, incinerated (500 °C, 4 h) and re-weighed to calculate the final ash-free dry mass (AFDM). Invertebrates were sorted, and litter-consuming detritivores were counted and identified under a binocular microscope to the highest taxonomic level possible (mostly species or genus, and family in some cases), using available literature and local expert knowledge.

**Calculation of variables**. We quantified litter decomposition in each litterbag as the proportion of litter mass loss (LML) per degree day (dd), to account for differences in temperature across sites; LML = [initial AFDM (g) – final AFDM (g)]/initial AFDM (g), where initial AFDM was previously corrected by leaching, drying and ash content, which were estimated in the laboratory[55]. We calculated detritivore-mediated decomposition as the difference in LML between paired coarse-mesh and fine-mesh litterbags[30]. Total and detritivore-mediated decomposition were strongly correlated ($r^2 = 0.90$, $p < 0.001$), but we used both as response variables in the analyses because the former is more relevant at the ecosystem level and the latter reflects patterns mediated solely by detritivores.

We quantified detritivore diversity in each coarse-mesh litterbag as taxon and family richness; as they were strongly correlated ($r^2 = 0.90$, $p < 0.0001$), we used family richness for analyses to avoid taxonomic inconsistencies among sites. We quantified abundance as the number of individuals per litterbag. We estimated total biomass based on mean body size using published equations for each family, and mean body size based on abundance and the mean of a body size category (2.5–5.0, 5.0–10.0, 10–20, 20–40 and 40–80 mm) that was assigned to each family using available literature[56–63].

**Data analyses**. We examined the influence of detritivore diversity, abundance, biomass, mean body size, latitude and the interactions between detritivore variables and latitude on decomposition, using generalised additive models (GAMs, gam function, 'mgcv' package v. 1.8.31[64,65]) and a model selection (dredge function,

'MuMIn' package v. 1.43.17) based on Akaike weights[66]. A model selection approach was used to identify which factors and interactions were included in the models with the highest conditional probabilities (i.e. Akaike weights; Supplementary Table 2). Models were fitted using tensor product interaction smooths (ti) with a normal or gamma distribution (depending on model fit and residuals) and the identity-link function[67]. We used this type of model instead of a linear model because preliminary data exploration showed the existence of non-linear patterns[68]. Total or detritivore-mediated decomposition was the response variable, and detritivore diversity, abundance, biomass, mean body size, absolute latitude and the interactions between detritivore variables and latitude were predictors, fitted as smooth terms. Exploring differences among litter mixtures was beyond the scope of this study (but see Boyero et al.[52], where litter diversity effects on decomposition were examined based on the same experiment described here), so we averaged values of different mixtures rather than including the mixture as a random factor in a generalised additive mixed model, which would be highly complex and would not converge when using interactions and variance functions (see below). Spatial correlation among sites was tested using the autocorrelation function (ACF) with residuals of the final model; all values were <1 as recommended by Zuur et al.[67]. Abundance and biomass data were log (x + 1)-transformed to avoid the disproportionate influence of outlying data observations on model estimates[68]. As interactions of detritivore variables with latitude were significant, we explored the relationships for tropical (≤23° of latitude), temperate (24–60°) and boreal zones (>60°) through a model that was similar to the one described above, but with latitude as a categorical rather than continuous predictor. This was done to facilitate the representation and interpretation of complex non-linear relationships between two continuous predictors.

We explored differences in detritivore variables across realms, biomes and climates with linear mixed-effects models (lme function, 'nlme' package v. 3.1.151[69]) where realm, biome or climate were fixed factors and litter mixture type was a random factor, followed by pairwise comparisons using adjusted P-values (glht and mcp functions, 'multcomp' package v. 1.4.13[70]). The variance was allowed to differ among realms and biomes using the VarIdent structure. Normalised residuals of the final model were inspected with plots of residuals vs. each predictor, and no pattern was observed. Variation in assemblage composition was explored with non-metric multidimensional scaling (NMDS, monoMDS function, 'vegan' package v. 2.5.6[71]) calculated on Hellinger transformed abundance data and permutational analysis of variance (PERMANOVA) based on a Bray–Curtis dissimilarity matrix. We compared realms, biomes and climates (adonis function, 'vegan' package), followed by pairwise comparisons (pairwise. adonis function), and determined which were the most representative families in each assemblage (simper function). All analyses were run on R v. 4.0.2.

**Reporting summary**. Further information on research design is available in the Nature Research Reporting Summary linked to this article.

## Data availability

Data supporting the findings of this study are available at https://doi.org/10.6084/m9.figshare.14245538.v1.

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

## Acknowledgements
We thank the many students and technicians who helped with research in different regions (S. Andrade, U. Apodaka, K. Barragán, A. J. Boulton, G. Diedericks, R. Roßberg, J. Rodger, M. Sachtleben, A. Tapia, A. Villarreal, V. Villarreal and others). This study was part of the DecoDiv project conducted by the GLoBE network (www.globenetwork.es), which is coordinated by L. B. Most research was based on crowdfunding (details on specific funding sources at each region are given in Supplementary Information). Project coordination was funded by Basque Government funds (Ref. IT951-16) to the Stream Ecology Group (UPV/EHU, Spain).

## Author contributions
The study was designed and coordinated by L.B., with help from N.L.-R., J.P. and R.G.P. All authors (mostly listed alphabetically: L.B., N.L.-R., A.M.T., J.P., F.C.-A., R.G.P., J.B., R.J.A., S.A., L.A.B., A.B., F.J.B., A.C., M.C., A.R.C., I.C.C., B.J.C., J.J.C., A.M.C.-S., E.C., S.C., C.C.C., A.C., A.M.D., M.D., E.S.D., M.E.D., M.M.D., A.C.E., R.F., A.S.F., T.F., E.A.G., G.G., P.E.G., M.O.G., J.E.G., S.G., J.F.G.J., M.A.S.G., D.C.G., R.O.H.J., N.H., C.H., D.I., T.I., S.K.K., A.L.-D., K.L., M.L., R.M., R.T.M., F.O.M., M.M., B.G.M., A.O.M., C.M.M., J.A.M., S.M., T.M., J.N.N., A.R., J.S.R., J.R., J.R.-R., J.M.S., R.S., F.S., A.S., N.S.D.T., S.D.T., J.R.T., M.V., A.W. and C.M.Y.) conducted research. Data management and analysis were performed by L.B., N.L.-R., AMT, J.P., and F.C.-A. The manuscript was written by L.B. with significant contributions from N.L.-R., J.P. and R.G.P. and feedback from the other authors. Figures were made by J.B.

## Competing interests
The authors declare no competing interests.

## Additional information

[1]Department of Plant Biology and Ecology, University of the Basque Country (UPV/EHU), Leioa, Spain. [2]IKERBASQUE, Bilbao, Spain. [3]Department of Ecology, University of Brasília (UnB), Brasília, Brazil. [4]Instituto Iberoamericano de Desarrollo Sostenible, Universidad Autonoma de Chile, Temuco, Chile. [5]Centre for Tropical Water and Aquatic Ecosystem Research (TropWATER), James Cook University, Townsville, QLD, Australia. [6]College of Science and Engineering, James Cook University, Townsville, QLD, Australia. [7]Research Unit of Biodiversity (CSIC, UO, PA), Oviedo University, Mieres, Spain. [8]Museo Nacional de Ciencias Naturales-CSIC, Madrid, Spain. [9]INIBIOMA (Universidad Nacional del Comahue - CONICET), Bariloche, Argentina. [10]Government Arts College, Melur, Madura, Tamil Nadu, India. [11]Biological Sciences, School of Natural Sciences,

University of Tasmania, Hobart, TAS, Australia. [12]Department of Aquatic Sciences and Assessment, Swedish University of Agricultural Sciences, Uppsala, Sweden. [13]Department of Ecology, Federal University of Rio Grande do Norte (UFRN), Rio Grande do Norte, Brazil. [14]Instituto de Ciências Biológicas, Universidade Federal de Minas Gerais, Belo Horizonte, Minas Gerais, Brazil. [15]Instituto de Biologia, Universidade Federal da Bahia, Bahia, Brazil. [16]Rhithroecology Pty Ltd., Blackburn, VIC, Australia. [17]Department of Ecosystem Science and Management, Penn State University, University Park, PA, USA. [18]Department of Biology and Geology, University of Almería, Almería, Spain. [19]Centro para la Investigación en Sistemas Sostenibles de Producción Agropecuaria (CIPAV), Cali, Colombia. [20]Illinois River Biological Station, University of Illinois Urbana-Champaign, Havana, IL, USA. [21]Laboratoire Écologie Fonctionnelle et Environnement, Université de Toulouse, CNRS, Toulouse, France. [22]Faculty of Tourism and Leisure, University of Physical Education, Kraków, Poland. [23]Department of Biology, Georgia Southern University, Statesboro, GA, USA. [24]Freshwater Macroinvertebrate Laboratory Gorgas Memorial Institute for Health Studies (COZEM-ICGES), Panama City, Panama. [25]Department of Experimental Limnology, Leibniz Institute of Freshwater Ecology and Inland Fisheries (IGB), Stechlin, Germany. [26]Graduate Program in Ecology, Federal University of Rio Grande do Norte (UFRN), Natal, Brazil. [27]Departamento de Ciencias Ambientales, Universidad Católica de Temuco, Temuco, Chile. [28]Facultad de Ciencias Ambientales y Centro EULA-Chile, Universidad de Concepción, Concepción, Chile. [29]School of Biological Sciences, The University of Western Australia, Crawley, WA, Australia. [30]Instituto BIOSFERA, Universidad San Francisco de Quito, Quito, Ecuador. [31]Department of Ecology and Evolutionary Biology, Cornell University, Ithaca, NY, USA. [32]Institute of Nature Conservation, Polish Academy of Sciences, Kraków, Poland. [33]Research Institute for the Environment and Livelihoods, Charles Darwin University, Casuarina, NT, Australia. [34]Water Laboratory and Physicochemical Services (LASEF), Autonomous University of Chiriqui, David City, Panama. [35]Escuela de Biología, Universidad de San Carlos de Guatemala, Guatemala City, Guatemala. [36]Organismal Biology, Ecology and Evolution (OBEE) program, University of Montana, Montana, MO, USA. [37]Berlin Institute of Technology (TU Berlin), Berlin, Germany. [38]Departamento de Ciencias Ambientales, Universidad de Puerto Rico, San Juan, Puerto Rico. [39]Department of Life Sciences and Marine and Environmental Sciences Centre (MARE), University of Coimbra, Coimbra, Portugal. [40]Biometric Research, South Fremantle, WA, Australia. [41]Flathead Lake Biological Station, University of Montana, Polson, MT, USA. [42]Instituto Nacional de Pesquisas da Amazônia–INPA, Coordenação de Biodiversidade–COBIO, Manaus, Amazonas, Brazil. [43]Department of Mathematical Sciences, Stellenbosch University, Matieland, South Africa. [44]Biodiversity Informatics Unit, African Institute for Mathematical Sciences, Cape Town, South Africa. [45]Integrated Graduate School of Medicine, Engineering, and Agricultural Sciences, University of Yamanashi, Kofu, Japan. [46]Faculty of Life and Environmental Sciences, University of Yamanashi, Kofu, Japan. [47]Egerton University, Egerton, Kenya. [48]Laboratorio de Contaminación Acuática y Ecología Fluvial, Universidad del Zulia, Maracaibo, Venezuela. [49]Department of Entomology, Museums Victoria, Melbourne, VIC, Australia. [50]Department of Fisheries and Aquatic Science, University of Eldoret, Eldoret, Kenya. [51]Department of Biological Sciences, Oakland University, Rochester, MI, USA. [52]INRAE, UR-RiverLy, Centre de Lyon-Villeurbanne, Villeurbanne Cedex, France. [53]Faculty of Environmental Earth Science, Hokkaido University, Sapporo, Hokkaido, Japan. [54]Department of Applied Ecology, North Carolina State University, Raleigh, NC, USA. [55]Department of Forest and Conservation Sciences, University of British Columbia, Vancouver, BC, Canada. [56]Departamento de Ecologia, Universidade Federal de Goiás (UFG), Goiânia, Goiás, Brazil. [57]Australian Rivers Institute, Griffith University, Nathan, QLD, Australia. [58]Université Julius N'Yerere de Kankan, Kankan, Guinea. [59]School of Science, Technology and Engineering, University of the Sunshine Coast, Sunshine Coast, QLD, Australia. ✉email: luz.boyero@ehu.eus

