## [Peer Review File · Nature Communications]

Reviewer comments, first round

Reviewer #1 (Remarks to the Author):

Dear authors and Editor,

I have now read the manuscript entitled "Impacts of detritivore extinctions on instream decomposition are the greatest in the tropics" by Boyero et al. Here, the authors have performed a very ambitious standardized study on the global scale, encompassing different types of habitats, biomes, and continents, in an attempt to answer the question whether detritivore taxonomic richness drives litter decomposition rates in streams, and if such an effect varies across biomes.

I found the paper very well written, and the results are indeed interesting. However, I have some comments and queries, especially when it comes to the uneven design (division into biomes) and to some of the very general statements in the discussion. I list these queries in the order in which they appear throughout the manuscript.

Title: The title is overstated. The authors did not investigate extinctions, but rather how variation in taxonomic richness might explain decomposition rate.

Line 210: Here, the authors suggest that climate change, i.e. increasing temperatures, may render tropical stream detritivores more vulnerable for extinction than stream detritivores in other biomes (and they later go on to state this more as a fact on Line 244). However, I wonder how much of a change in water temperature the comparatively small expected increase in temperature due to warming would cause, and how important such an increase would be compared to, e.g., increased risk of prolonged droughts in especially temperate, but maybe also boreal, regions? Hence, I think the authors are a bit selective in how they discuss the general relevance of their results, and would appreciate a bit more nuanced discussion.

Line 225: The authors state that smaller detritivores are more sensitive to stressors than larger ones (and includes a reference). Is this a general effect of size across all taxa, or is this difference due to size because some large taxa (e.g. caddisflies) are more tolerant than smaller taxa (e.g. chironomids)? Knowing that there often is a large difference in sensitivity among taxa, I find it hard to see that size per se would be more important than taxonomic differences (disregarding size). Further, I know of at least some examples where very small taxa (e.g. chironomids, among them some detritivores) are among the most tolerant taxa and large taxa (e.g. caddisflies) can be very sensitive. Moreover, considering increasing temperature as a stressor, previous studies have suggested that one response would be decreases in size, which goes against the arguments of the authors in this paper. Hence, this text, and the text about climate effects, becomes very speculative (and subjective), I think. The authors should present a more nuanced view.

Line 262: I find the description of which litters were used a bit hard to understand. These were litters collected in a few select locations (Table S1), and then distributed around the world? If so, how do the authors think that non-native litter could influence their results. Previous studies have shown litter type and origin matters, and that even intra-specific variation in litter quality may matter for decomposition rates. Hence, litter that is novel to stream detritivores may alter the results. I therefore think the authors need to explain this part better, and also later discuss potential implications of how (and what) litter was used.

Line 272: I would like to see also the variation around the mean mass loss, and it would be most informative to see how this differed among regions/biomes.

Line 292-293: I understand the use of coarser taxonomy, but could this have implications on the

results. For example, are families differently diverse in the different regions, i.e. could "diversity" be missed by grouping taxa within families, in some regions more so than in others, and therefore influence the relationship between richness and decomposition rates and consequently interpretations of the results?

Line 325: The division of the data into biomes is good, in theory, but this has (when looking at the figures) resulted in a substantial difference in data points among biomes. Especially, the boreal region has very few data points, and much fewer than the other biomes. Could this have prevented the authors from finding patterns in the boreal zone (which is one of the main conclusions)? Are the authors confident in that the results are comparable among biomes, given this large difference in data? The authors should at least discuss some potential implications of this shortcoming.

Reviewer #2 (Remarks to the Author):

This is a very well written paper that tests how detritivore insect diversity affects decomposition of leaf litter in streams. The major claim of this paper is that detritivore diversity increases decomposition of leaf litter in streams. A strength of this research is the geographic scope of the experiment, which spans 38 streams around the world. The research is important because leaf litter is a major energy source that fuels many stream ecosystems. Understanding how decreases in biodiversity affect this ecosystem process is important for carbon cycling. The work is of interest to Ecosystem ecologists and fits into the general debate about the role of diversity in driving ecosystem function. The methods are straightforward and standard. The analysis is appropriate. I think that it would be possible for others to reproduce the work. The description of the analysis is detailed enough that it could be replicated.

There are a few issues that the authors could address to strengthen the impact. The authors should discuss in more detail why there was a significant relationship between diversity of invertebrates and total decomposition but not in detritivore decomposition. They base their final analysis on the detritivore decomposition packs but total decomposition is what streams experience. There are a few possible reasons for the discrepancy. Multiple factors could account for discrepancy and I would like to hear what the authors think. Perhaps the small mesh large mesh technique didn't work very well if small insects were able to penetrate or maybe there was some interaction between microbes and detritivores small mesh bags. Given that detritivore diversity affected total decomposition, which is the most important variable, I would not be too quick to declare that detritivore diversity is more important in the tropics than in temperate zones. It appears that there are similar patterns in the significance of interaction terms when comparing full decomposition versus detritivore decomposition but given that full decomposition is not presented in detail it would be helpful and confirming to know that the directional patterns are similar. The description about the leaf type mixtures left me confused about what actually happened and needs to be clarified. I am assuming that packs were paired, the same litter was incubated in all and there were no interactions with leaf type * detritivore diversity and decomposition. The distribution of sample streams is concentrated in some areas and sparse in others (Africa, Russia). The authors should address how this might affect conclusions. I understand how logistical details and history of ecology biases towards some regions so I don't fault the authors for having a more random distribution of sites but nevertheless it needs to be addressed.

The authors also conclude that the distribution of aquatic detritivores is driven by biogeography based on the distribution of taxa. This is a very interesting point but it needs to be backed up with a phylogenetic analysis. Are there clades of insects that are only found in one of the regions versus the others? Although an interesting observation – it needs to be accompanied by a more robust comparison of different models. As written, the authors present a subset of pictures select taxa (which I like) but it is hard to see if these are cherry picked to support their result or if the entire data set supports this conclusion. It also looks like from the first MDS figure that the samples from Africa group with the Northern Hemisphere which seems inconsistent with their conclusion.

My main concern is the presentation of the results. The very small graphs – were hard to see even when zooming at 200%. Similarly, all of the abbreviations were difficult to follow. Spell out the names in the figure legends. It won't take much space but will make it easier to follow. Because the authors present the same results in multiple renditions (biome, realm) it becomes hard to follow the number of sample sizes. Perhaps there is another way to present the range of values in the bar graphs so that each panel can be bigger.

I think that this research has the potential to influence the field because it is a broad study that shows relationships between detritivore diversity and decomposition. The differences between the temperate and tropical streams is compelling although I don't understand why the authors put more weight on the derived values for detritivore effect than on total decomposition.

Jane C. Marks

COMMENTS FROM REVIEWER #1

I have now read the manuscript entitled “Impacts of detritivore extinctions on instream decomposition are the greatest in the tropics” by Boyero et al. Here, the authors have performed a very ambitious standardized study on the global scale, encompassing different types of habitats, biomes, and continents, in an attempt to answer the question whether detritivore taxonomic richness drives litter decomposition rates in streams, and if such an effect varies across biomes. I found the paper very well written, and the results are indeed interesting. However, I have some comments and queries, especially when it comes to the uneven design (division into biomes) and to some of the very general statements in the discussion. I list these queries in the order in which they appear throughout the manuscript.

Comment #2: The title is overstated. The authors did not investigate extinctions, but rather how variation in taxonomic richness might explain decomposition rate.

Response: We considered ‘extinctions’ to be a punchier synonym of ‘diversity loss’, but agree that the second term might be more appropriate and have replaced it.

Comment #3: Line 210: Here, the authors suggest that climate change, i.e. increasing temperatures, may render tropical stream detritivores more vulnerable for extinction than stream detritivores in other biomes (and they later go on to state this more as a fact on Line 244). However, I wonder how much of a change in water temperature the comparatively small expected increase in temperature due to warming would cause, and how important such an increase would be compared to, e.g., increased risk of prolonged droughts in especially temperate, but maybe also boreal, regions? Hence, I think the authors are a bit selective in how they discuss the general relevance of their results, and would appreciate a bit more nuanced discussion.

Response: We have replaced “climate change” by “climate warming” and added a reference about the importance of increased droughts at higher latitudes (L. 216-217).

Comment #4: Line 225: The authors state that smaller detritivores are more sensitive to stressors than larger ones (and includes a reference). Is this a general effect of size across all taxa, or is this difference due to size because some large taxa (e.g. caddisflies) are more tolerant than smaller taxa (e.g. chironomids)? Knowing that there often is a large difference in sensitivity among taxa, I find it hard to see that size per se would be more important than taxonomic differences (disregarding size). Further, I know of at least some examples where very small taxa (e.g. chironomids, among them some detritivores) are among the most tolerant taxa and large taxa (e.g. caddisflies) can be very sensitive. Moreover, considering increasing temperature as a stressor, previous studies have suggested that one response would be decreases in size, which goes against the arguments of the authors in this paper. Hence, this text, and the text about climate effects, becomes very speculative (and subjective), I think the authors should present a more nuanced view.

Response: We agree and have rewritten this part to address the comment (L. 229-230).

Comment #5: Line 262: I find the description of which litters were used a bit hard to understand. These were litters collected in a few select locations (Table S1), and then distributed around the world? If so, how do the authors think that non-native litter could influence their results. Previous studies have shown litter type and origin matters, and that even intra-specific variation in litter quality may matter for decomposition rates. Hence, litter that is novel to stream detritivores may alter the results. I therefore think the authors need to explain this part better, and also later discuss potential implications of how (and what) litter was used.

Response: We have extended the explanation of the litter mixtures used. We explain that our main interest here was to cover a wide range of plants and mixtures across a broad range of functional diversity and thus increase the generality of our results (L. 271-278). Each species was indeed collected at a different

location and distributed among partners; we now acknowledge that home-field-advantage effect is possible, but we believe it to be negligible based on existing literature for instream decomposition (Fenoy et al. 2016 FEMS Microbiol Eco. 92, f1w169, Fugère et al. 2020 Freshwa. Sci 497–507). Terrestrial studies have also shown that home-field advantage explains much lower variability in decomposition than other factors such as litter traits and climate (Ayers et al. 2009 Soil Bio Biochem 41, 606–610; Makkonen et al. 2012 Ecol Lett 15: 1033–1041).

Comment #6: Line 272: I would like to see also the variation around the mean mass loss, and it would be most informative to see how this differed among regions/biomes.

Response: Standard deviations have been now added to these values (L. 290-291), as well as a supplementary figure showing values for each biome (Fig. S1).

Comment #7: Line 292-293: I understand the use of coarser taxonomy, but could this have implications on the results. For example, are families differently diverse in the different regions, i.e. could “diversity” be missed by grouping taxa within families, in some regions more so than in others, and therefore influence the relationship between richness and decomposition rates and consequently interpretations of the results?

Response: Most specimens were identified to the genus or species level, but some were identified to family (now indicated; L. 197-298), due to differences in taxonomic knowledge. In consequence, we could only choose between using taxon richness or family richness (which were strongly related; $r^2 = 0.90$, $p < 0.0001$), and not species richness, which was unavailable. We chose to use family richness for consistency across regions. We now nevertheless mention this issue in the discussion (L. 197-199).

Comment #8: Line 325: The division of the data into biomes is good, in theory, but this has (when looking at the figures) resulted in a substantial difference in data points among biomes. Especially, the boreal region has very few data points, and much fewer than the other biomes. Could this have prevented the authors from finding patterns in the boreal zone (which is one of the main conclusions)? Are the authors confident in that the results are comparable among biomes, given this large difference in data? The authors should at least discuss some potential implications of this shortcoming.

Response: We now mention this issue in the discussion (L. 202-203). However, the lower representation of the boreal climate in our dataset did not preclude finding strong influences of detritivore abundance and biomass (but not diversity) in this climatic zone (Fig. 2).

COMMENTS FROM REVIEWER #2

This is a very well written paper that tests how detritivore insect diversity affects decomposition of leaf litter in streams. The major claim of this paper is that detritivore diversity increases decomposition of leaf litter in streams. A strength of this research is the geographic scope of the experiment, which spans 38 streams around the world. The research is important because leaf litter is a major energy source that fuels many stream ecosystems. Understanding how decreases in biodiversity affect this ecosystem process is important for carbon cycling. The work is of interest to Ecosystem ecologists and fits into the general debate about the role of diversity in driving ecosystem function. The methods are straightforward and standard. The analysis is appropriate. I think that it would be possible for others to reproduce the work. The description of the analysis is detailed enough that it could be replicated.

Comment #9: There are a few issues that the authors could address to strengthen the impact. The authors should discuss in more detail why there was a significant relationship between diversity of invertebrates and total decomposition but not in detritivore decomposition. They base their final analysis on the detritivore decomposition packs but total decomposition is what streams experience. There are a few possible reasons for the discrepancy. Multiple factors could account for discrepancy and I would like to hear what the authors think. Perhaps the small mesh large mesh technique didn't work very well if small insects were able to penetrate or maybe there was some interaction between microbes and detritivores small mesh bags. Given that detritivore diversity affected total decomposition, which is the most important variable, I would not be too quick to declare that detritivore diversity is more important in the tropics than in temperate zones. It appears that there are similar patterns in the significance of interaction terms when comparing full decomposition versus detritivore decomposition but given that full decomposition is not presented in detail it would be helpful and confirming to know that the directional patterns are similar. The description about the leaf type mixtures left me confused about what actually happened and needs to be clarified. I am assuming that packs were paired, the same litter was incubated in all and there were no interactions with leaf type * detritivore diversity and decomposition.

Response: Coarse-mesh and fine-mesh litterbags were indeed paired, and they both contained the same amount and type of litter, as indicated now in the text (L. 279-280). We agree that total decomposition is more relevant at the ecosystem scale than detritivore-mediated decomposition, the latter indicating detritivore performance; we explored both variables to give greater insight into the process, as explained in the manuscript (L. 307-309). We, however, do not believe that we focused our discussion unduly on detritivore-mediated decomposition. The key result of our analysis is the significant interactions between decomposition and detritivore variables, and this result was exactly the same for both decomposition variables; our discussion thus refers to both (see also Fig. 2). When there is a significant interaction between two factors, this overrides the significance of each factor separately (Vargas et al. 2018 Analysis and interpretation of interactions of fixed and random effects, in Applied Statistics in Agricultural, Biological, and Environmental Sciences, Chapter 7; Quinn & Keough 2002 Experimental design and data analysis for biologists. Cambridge university press), so the interpretation of the main effect of detritivore diversity on total decomposition is difficult and we could only speculate about it.

Comment #10: The distribution of sample streams is concentrated in some areas and sparse in others (Africa, Russia). The authors should address how this might affect conclusions. I understand how logistical details and history of ecology biases towards some regions so I don't fault the authors for having a more random distribution of sites but nevertheless it needs to be addressed.

Response: This is now mentioned in the manuscript (L.259-261).

Comment #11: The authors also conclude that the distribution of aquatic detritivores is driven by biogeography based on the distribution of taxa. This is a very interesting point but it needs to be backed up with a phylogenetic analysis. Are there clades of insects that are only found in one of the regions versus the others? Although an interesting observation – it needs to be accompanied by a more robust comparison of different models. As written, the authors present a subset of pictures select taxa (which I like) but it is hard to see if these are cherry picked to support their result or if the entire data set supports this conclusion. It also looks like from the first MDS figure that the samples from Africa group with the Northern Hemisphere which seems inconsistent with their conclusion.

Response: There were 26 families showing a Laurasian distribution and 14 showing a Gondwanan distribution (L. 234-235, Table S4). Even in the absence of formal phylogenetic analysis (which we find very interesting but outside the scope of this paper; see also our response to the Editor's comment above), we believe that these distribution patterns suggest a key role of biogeography, as discussed in the manuscript. We now mention the absence of phylogenetic analysis in our study (L. 236-237). The Afrotropical realm was grouped with the Neotropical and Australasian realms in the NMDS, but the low size of the previous image made it difficult to see it; we apologize and hope that the new split figure offers better clarity.

Comment #12: My main concern is the presentation of the results. The very small graphs – were hard to see even when zooming at 200%. Similarly, all of the abbreviations were difficult to follow. Spell out the names in the figure legends. It won't take much space but will make it easier to follow. Because the authors present the same results in multiple renditions (biome, realm) it becomes hard to follow the number of sample sizes. Perhaps there is another way to present the range of values in the bar graphs so that each panel can be bigger.

Response: We have split Fig. 1 in three different figures, in order to improve the size and clarity of the graphs, and have spelled out the names in the corresponding legends.

Comment #13: I think that this research has the potential to influence the field because it is a broad study that shows relationships between detritivore diversity and decomposition. The differences between the temperate and tropical streams is compelling although I don't understand why the authors put more weight on the derived values for detritivore effect than on total decomposition.

Response: We appreciate the positive and constructive comments. We disagree only regarding emphasis on detritivore-mediated decomposition, as explained in response to comment #9 above.

Reviewer comments, second round –

[Editor's note: both reviewers found their concerns to have been addressed satisfactorily and did not have any remaining comments to the authors.]